# Impact of digital breast tomosynthesis on screening performance and interval cancer rates compared to digital mammography: A meta-analysis

**Xuewen Liu, Ting Yang, Juan Yao** [iD] *

The First Affiliated Hospital of Xinjiang Medical University, Ürümqi, China

\* 13139929005@163.com

**Data Availability Statement:** All relevant data are within the manuscript and its Supporting Information files.

**Funding:** The author(s) received no specific funding for this work.

## Abstract

### Background

The performance of digital breast tomosynthesis (DBT) alone, digital mammography (DM) plus DBT, and synthesized mammography (SM) plus DBT, in comparison to DM in breast cancer screening, remains a topic of ongoing debate. The effectiveness of these modalities in reducing interval cancer rates (ICR) is particularly contentious.

### Materials and methods

A database of data was searched for articles published until July 2024. Initially, the pooled sensitivity and specificity of DBT (DBT alone, DM/DBT, and SM/DBT) and DM were estimated. Additionally, the sensitivity of breast cancer screening and ICR for DBT alone, DM/DBT, and SM/DBT compared to DM. The characteristics of interval breast cancer were compared with those screening BC, alongside differences across various screening methods.

### Results

Eleven studies comparing DBT and DM were included. The sensitivity of DBT was higher than that of DM, with rates of 86% (95%CI: 81, 90) and 80% (95%CI: 76, 84), respectively. The specificities of both modalities were similar, recorded at 96% (95%CI: 95, 98) and 96% (95%CI: 95, 97), respectively. In comparison to DM, the screening sensitivities of DBT, DM/DBT, and SM/DBT were increased by 4.33% (95% CI: 1.52, 7.13), 6.29% (95% CI: 2.55, 10.03), and 5.22% (95% CI: 1.35, 9.10), respectively; however, the difference in the ICR was not statistically significant.

### Conclusion

DBT offers advantages in enhancing the sensitivity of breast cancer screening; however, its impact on ICR remains uncertain. Consequently, further research is necessary to comprehensively evaluate both the effectiveness of screening and the potential risks associated with DBT.

**Competing interests:** The authors have declared that no competing interests exist.

## 1. Introduction

Breast cancer(BC) is one of the most prevalent cancers worldwide. According to the latest report from the American Society of Clinical Oncology (ASCO), the incidence of female BC is projected to continue its gradual rise, increasing by approximately 0.6% per year through 2024 [1]. ASCO's 2022 report emphasizes that early diagnosis of BC can reduce the risk of death by 30–40% [2]. Furthermore, BC identified during early screening are generally less invasive and tend to exhibit slower growth. Consequently, BC screening is a crucial strategy for the early detection of the disease and for improving survival rates among cancer patients. The World Health Organization (WHO) recommends that countries implement appropriate BC screening programs to mitigate mortality and decrease the incidence of late-stage cancer [3].

While screening can reduce breast cancer mortality, current evidence is insufficient to demonstrate that various screening methods significantly lower breast cancer mortality rates [4]. This limitation may stem from the fact that assessing breast cancer mortality as an endpoint necessitates large-scale trials and extended follow-up periods. Consequently, some studies have proposed using the ICR as a crucial indicator for evaluating the effectiveness of BC screening methods [5]. The ICR represents the proportion of BC cases that exhibit clinical progression during regular screening intervals, thereby serving as an indirect measure of the screening method's effectiveness.

Interval BC often characterized as a false-negative diagnosis [6], refers to BC identified within 6 to 24 months following a negative screening result or within 6 months of a positive screening result. Surveys indicate that the histopathological features of interval BC tend to be more invasive [7,8]. The effectiveness of screening modalities should be evaluated not only based on the types of detected cancers but also on those that remain undetected. Understanding ICR and histopathological characteristics is essential for assessing the performance of BC screening modalities.

BC screening guidelines recommend DM as the primary screening modality for women. However, DM has a significant limitation: potential tissue overlap [9], particularly in women with dense breasts. Cancers that exhibit a density similar to that of the surrounding tissue may be easily missed, which contributes to interval BC being the most common omission anomaly [10]. Numerous studies have demonstrated that DBT can effectively reduce tissue superposition [11–13]. In the United States, DBT is increasingly replacing DM as the preferred imaging examination for routine screening [14]. For instance, a substantial reduction in ICR using DBT for screening was reported in the Malmö trial [15]. Furthermore, an Italian randomized controlled trial found that DBT combined with DM detects 70% more breast cancers than DM [16]. These findings may explain why DBT is gradually supplanting DM. However, in Europe, DBT is not yet utilized as a routine screening tool, possibly due to the absence of significantly lower ICR results, which raises concerns about the potential for overdiagnosis of BC [17].

To enhance the diagnostic efficiency of BC, combined diagnostic methods have garnered increasing attention. Research indicates that, compared to DM alone, DM/DBT [18,19] and SM/DBT [20,21], can significantly elevate the cancer detection rate. A study conducted during the Cordoba Breast DBT Screening Trial revealed that patients screened using DM/DBT experienced lower ICR than those screened with DM alone [22]. Additionally, a prospective study demonstrated that while SM/DBT enhanced screening sensitivity, it did not result in a significant reduction in ICR compared to DM alone [23].

As new screening methods for BC continue to emerge, a comprehensive meta-analysis is essential to evaluate the impact of DBT (DBT alone, DM/DBT, and SM/DBT) on ICR and the characteristics of interval BC. This analysis aims to elucidate the screening performance of DBT by comparing its sensitivity with DM alone in BC screening, while also examining ICR

during follow-up and the characteristics of interval BC. The purpose of this review is to propose new screening options for early BC detection and to advocate for a transition from DM screening alone to DBT or in combination with its modalities in future BC screening programs.

## 2. Materials and methods

### 2.1. Protocol and registration

This study followed the Preferred Reporting Items for Systematic Reviews and Meta-analyses (PRISMA) guidelines for reporting (2020-version) [24]. The protocol for this meta-analysis was registered in PROSPERO (CRD42024575432).

### 2.2. Data sources and searches

We conducted a search of PubMed, Web of Science, and the Cochrane Library for articles published up until July 2024 to identify eligible studies. Key terms searched included "Mammography," "Tomosynthesis," "Mammographies," "Digital Breast Tomosynthesis," and "interval breast cancer." The complete search strategy is available in 'S1 File.'

### 2.3. Selection of studies

Inclusion criteria: Comparative studies were included, including DBT alone versus DM, DM/DBT versus DM alone, and SM/DBT versus DM alone. There is sufficient information available to differentiate between true positives, false positives, false negatives (interval BC), and true negatives, thereby allowing for the determination of sensitivity and specificity. BC is identified through biopsy and/or follow-up. Exclusion criteria: Studies that do not involve population screening or initial screening populations. Incomplete relevant data, including a lack of cancer examination or interval BC data. Incomplete and unpublished research reports. Any disagreements or uncertainties were addressed through discussions with a third researcher, leading to a consensus.

### 2.4. Data extraction

Basic study information was extracted, including age, breast density grade, study design, and reference test (biopsy or follow-up). True positive, false positive, false negative, and true negative data were extracted to construct a 2×2 contingency table, which was then used to calculate the sensitivity and specificity of the screening technology. Additionally, the number and characteristics of interval BC were extracted, including histological type, pathological tumor size (pT) classification and the five molecular subtypes of invasive BC: Luminal A (LumA), Luminal B-/HER2-positive (LumB/HER2+), Luminal B-/HER2-negative-like (LumB/HER2-), HER2-type (HER2), triple-negative (TN), the grade (Grade) and distribution of lymph node status were also recorded. All statistical tests were conducted as two-sided. The primary endpoints included the incidence of ICR per 10,000 screens and the sensitivity of BC screening. The secondary endpoints focused on interval BC histopathological characteristics.

### 2.5. Statistical analysis

The evaluation will be conducted according to the criteria specified by the Quality Assessment of Diagnostic Accuracy Studies 2 (QUADAS-2) [25] tool for test accuracy studies. Summary statistics for sensitivity and specificity were estimated using Stata's bivariate random effects model. To further illustrate the diagnostic value, the summary receiver operating characteristic curve (SROC) was analyzed to calculate the area under the summary curve (AUC). To evaluate

study heterogeneity, defined as the variability in diagnostic accuracy across the primary studies, we employed the Higgins $I^2$ statistic. Heterogeneity levels will be classified as low, moderate, and high, corresponding to values of 25%, 50%, and 75%, respectively. Two univariate meta-regression analyses were performed to evaluate the intervention (DBT alone or in combination with DBT) and study type (prospective or retrospective) to assess sources of heterogeneity. Utilize the 'metan' command in Stata to calculate summary estimates for comparing the screening sensitivities and ICR of various screening tests. The results are summarized as risk differences (RDs) along with their corresponding 95% confidence intervals (CIs). The characteristics of interval BC were summarized for each study group, employing the chi-square test for categorical variables and the Kruskal-Wallis test for continuous variables.

## 3. Results

### 3.1. Literature search and study characteristics

A systematic literature search identified 662 documents. After excluding duplicates and irrelevant literature (**Fig 1**), 11 studies (n = 762,513) were ultimately included, encompassing

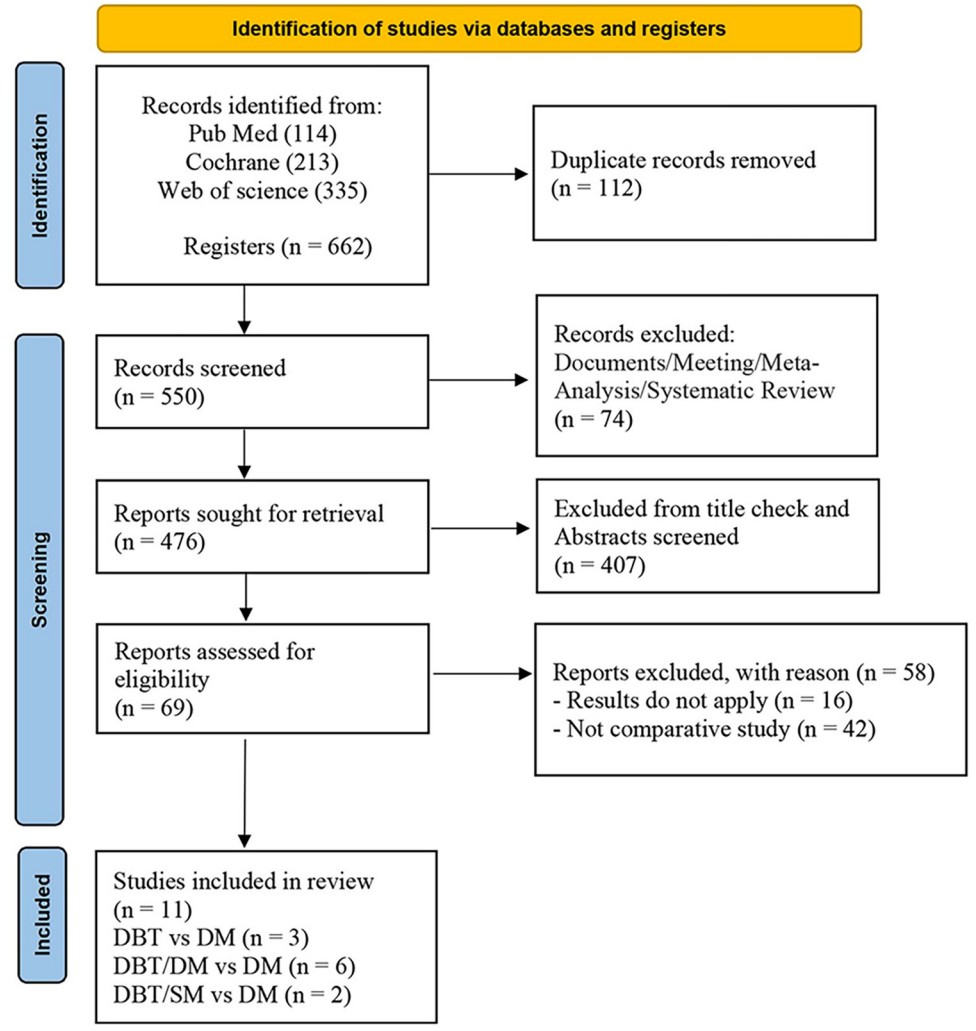

**Fig 1. Flow diagram of the study selection process.**

349,304 patients who received DBT (including DBT alone, DM/DBT, and SM/DBT). Three studies compared DBT alone to DM alone [26–28], while six studies evaluated DM/DBT against DM alone [16,22,29–32]. Additionally, two studies compared SM/DBT to DM alone [23,33]. Among the included studies, eight were prospective trials, while the remaining three were retrospective trials. Notably, only three of the 11 studies were randomized controlled trials (RCTs). The characteristics of the individual studies are summarized in (S1 Table).

### 3.2. Risk of bias

The risk of bias was rated as low for included studies after assessing by the modified QUADAS-2 checklist. The detailed risk of bias assessment is available in (S1 Fig).

### 3.3. Analysis of diagnostic accuracy of DBT

Among the 349,304 participants who underwent DBT screening (defined as DBT alone, DM/DBT, and SM/DBT), 2,511 cases of screened BC were detected. A bivariate random effects meta-analysis demonstrated that the pooled sensitivity of DBT was 86% (95% CI: 81, 90, $I^2 = 93.08\%$), while the pooled specificity was 96% (95% CI: 95, 98, $I^2 = 99.91\%$) (Fig 2A). In contrast, among the 413,209 participants screened for DM, 2,121 cases of screened BC were detected, with a pooled sensitivity of DM at 80% (95% CI: 76, 84, $I^2 = 85.88\%$) and a pooled specificity of 96% (95% CI: 95, 97, $I^2 = 99.93\%$) (Fig 2B). The data analysis depicted indicates that DBT demonstrates superior sensitivity when compared to traditional DM while maintaining equivalent specificity. This finding highlights the significant potential of DBT to improve BC detection.

The summary receiver operating characteristic (SROC) curves for DBT (Fig 3A) and DM alone (Fig 3B). The areas under the summary curves were 0.97 (95% CI: 0.95, 0.98) for DBT and 0.95 (95% CI: 0.92, 0.96) for DM alone, indicating that DBT has superior BC screening value, further affirming the screening advantages of DBT. Furthermore, the likelihood ratio dot plot presented (Fig 3C) that the convergence point falls within the (LRP>10, LRN>0.1) quadrant, suggesting that DBT possesses a high identification value for BC but limited exclusion capability. This emphasizes both the practicality of DBT in BC screening and the necessity for complementary screening strategies to enhance effective disease exclusion.

There was considerable heterogeneity in the pooled sensitivity and specificity of both the DBT and DM alone. To further elucidate the origins of heterogeneity within the study, two univariate meta-regression analyses were performed. The findings, detailed in (S2 Table), indicated that the mode of intervention (DBT alone or in combination with DBT) and study type (prospective or retrospective), could significantly influence heterogeneity (p<0.001). The Deeks funnel plot demonstrated that the studies were symmetrically distributed around the fitted line, providing no direct evidence of publication bias (p = 0.18) (refer to Fig 4).

In conclusion, the inclusion of DBT (DBT alone, DM/DBT, and SM/DBT) into BC screening protocols—substantially enhances the efficacy of the screening process. This methodology demonstrates superior performance compared to the exclusive use of DM screening, thereby facilitating earlier diagnoses, which may result in improved patient outcomes in the realm of BC screening.

### 3.4. Sensitivity analysis of breast cancer screening by DBT alone, DM/DBT, SM/DBT compared with DM

The sensitivity of screening with DBT (DBT alone, DM/DBT, and SM/DBT) was assessed in comparison to DM alone. The pooled risk difference RD = 5.24% (95% CI: 3.26, 7.22, $I^2 = 0\%$) demonstrated that the DBT screening group had a higher sensitivity compared to DM alone. (Fig 5) To compare various interventions, three studies were included that compared digital

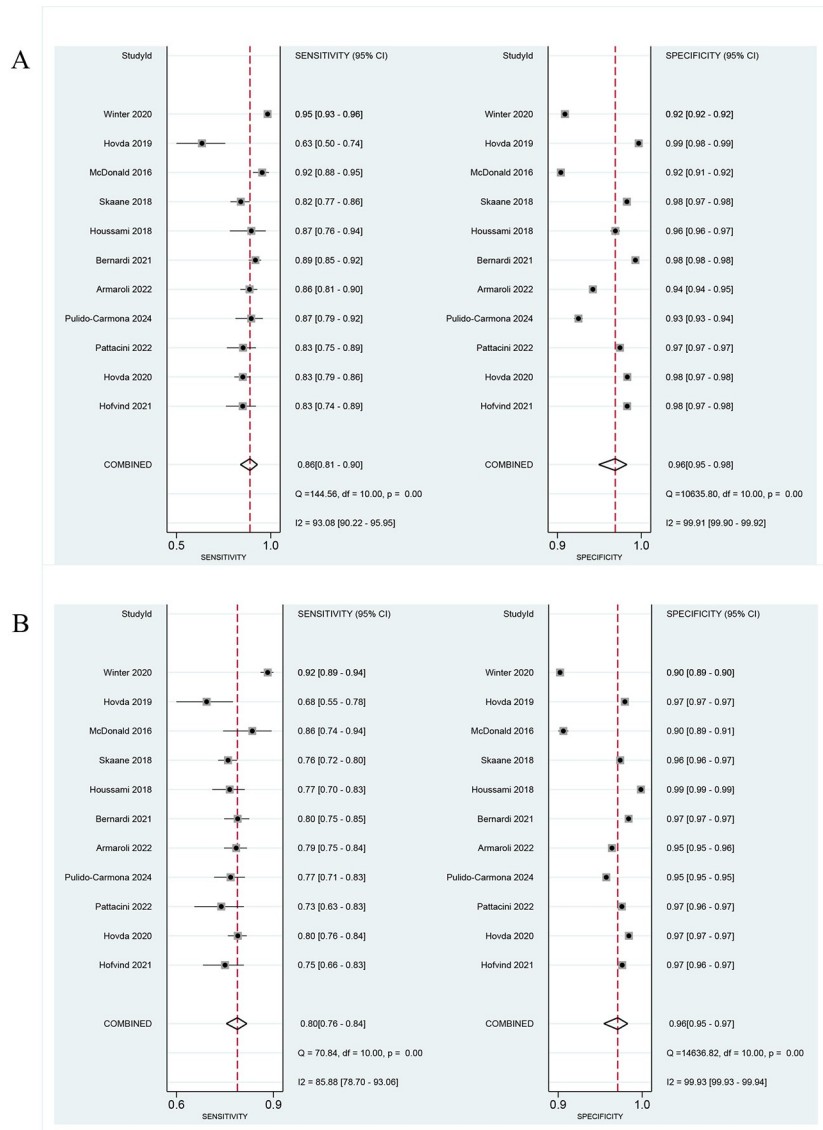

**Fig 2. Coupled forest plots of the pooled sensitivity and specificity of DBT and DM for breast cancer diagnosis.**
(A) Forest plots for DBT. (B) Forest plot for DM. The black square boxes denote either sensitivity (left panel) or specificity (right panel), and horizontal lines represent 95% CI for each study. The vertical dotted line indicates pooled summary estimates of sensitivity or specificity, and the diamond at the bottom indicates the 95% CI. Heterogeneity statistics ($I^2$ value, Q value) for sensitivity and specificity are displayed.

DBT alone with DM screening alone, encompassing a total of 339,465 subjects and detecting 1,806 cases of screening BC. Additionally, six comparative studies of DM/DBT versus DM screening were included, involving a total of 246,865 subjects and identifying 1,540 cases of screening BC. Furthermore, two comparative studies of SM/DBT versus DM screening were included, with a total of 176,183 participants and 1,286 cases of screening BC detected.

The study found that DBT alone exhibits a different sensitivity for BC screening compared to DM alone RD = 4.33% (95% CI: 1.52, 7.13, $I^2$ = 12.6%<25%), DBT alone can enhance sensitivity. There is a notable difference between the use of DM/DBT and DM alone RD = 6.29% (95% CI: 2.55, 10.03, $I^2$ = 0%). The DM/DBT screening enhances the sensitivity of BC detection compared to DM screening alone. Similarly, there is a difference between the use of screen

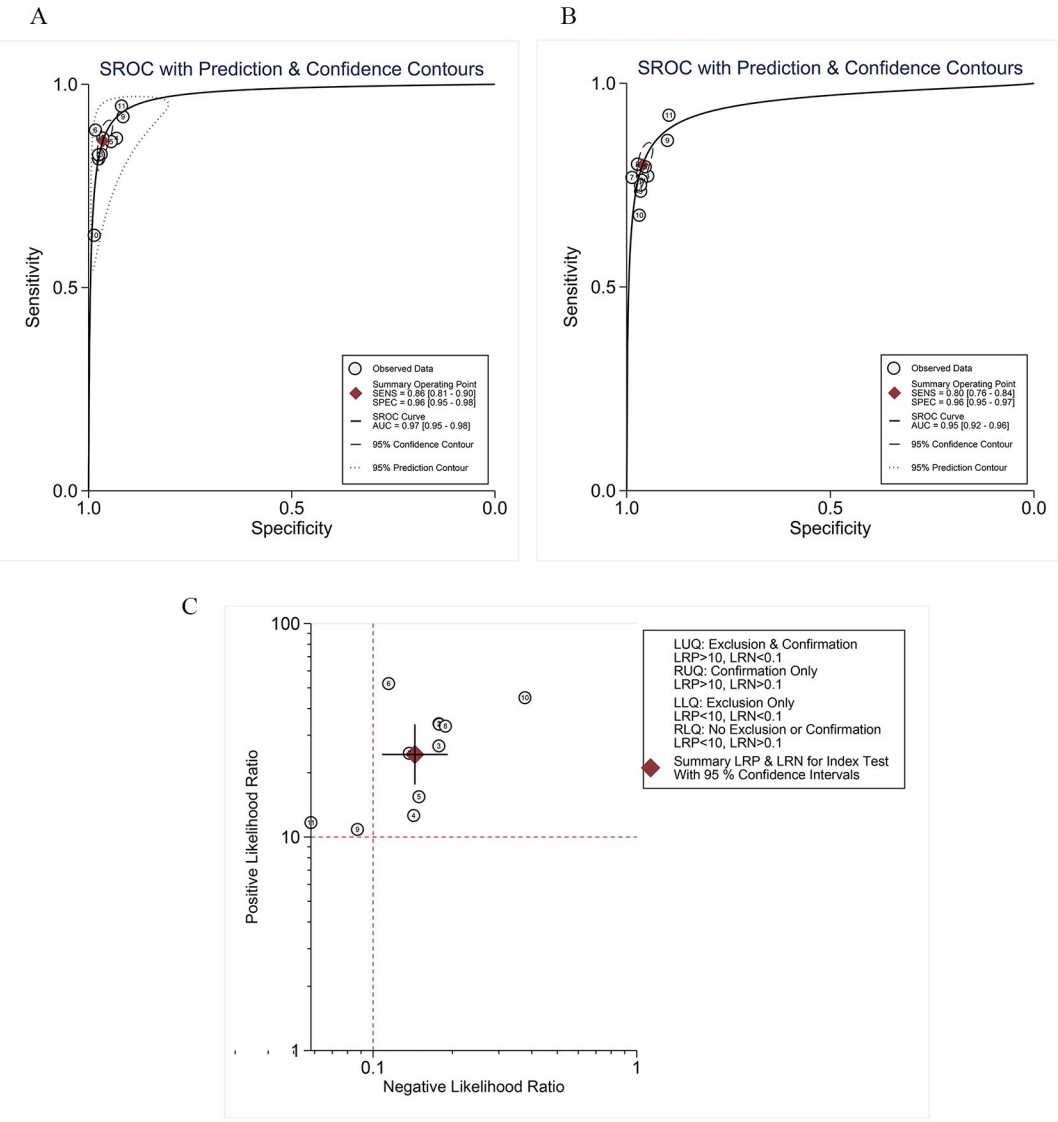

**Fig 3. The SROC curves for breast cancer diagnosis and the likelihood quadrant distribution plot.** (A) displays the SROC curves for DBT in breast cancer diagnosis, and (B) shows the SROC curves for DM. SROC curves show the individual (circles) and pooled (red square) sensitivity and specificity, and the dimension of each circle indicates the weight from the study sample size. (C) features the DBT likelihood quadrant distribution plot. LRP = likelihood ratio positive, LRN = likelihood ratio negative.

SM/DBT and DM alone RD = 5.22% (95% CI: 1.35, 9.10, $I^2$ = 58.5%<75%). The SM/DBT also increases sensitivity compared to DM screening alone.

In conclusion, a comparative analysis reveals that DBT used independently, as well as in conjunction with DM or SM exhibits significantly higher screening sensitivity than traditional DM screening conducted in isolation. These results indicate that integrating DBT with other mammographic techniques may enhance the early detection of BC.

### 3.5. Effect of DBT Alone, DM/DBT, and SM/DBT on ICR Compared with DM

Among the 349,304 participants who underwent DBT (DBT alone, DM/DBT, and SM/DBT), 360 cases of interval BC were detected. In contrast, among the 413,209 participants screened

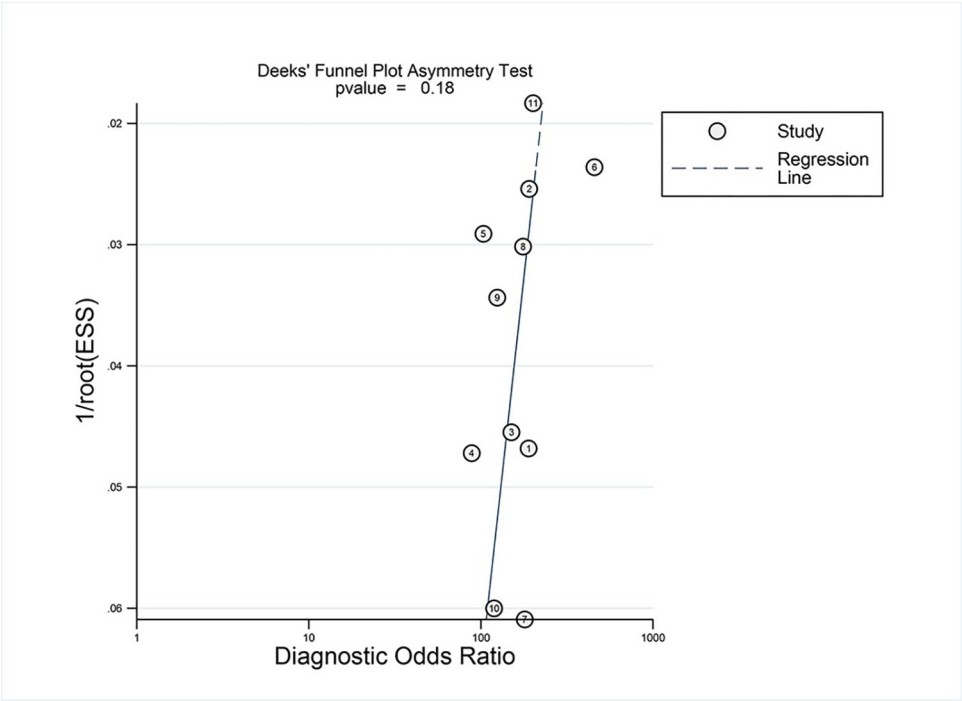

**Fig 4. Deeks' funnel plot for digital breast tomosynthesis.** The p-value of 0.18 for the slope coefficient indicates symmetry in the data and a low likelihood of publication bias. ESS effective sample size.

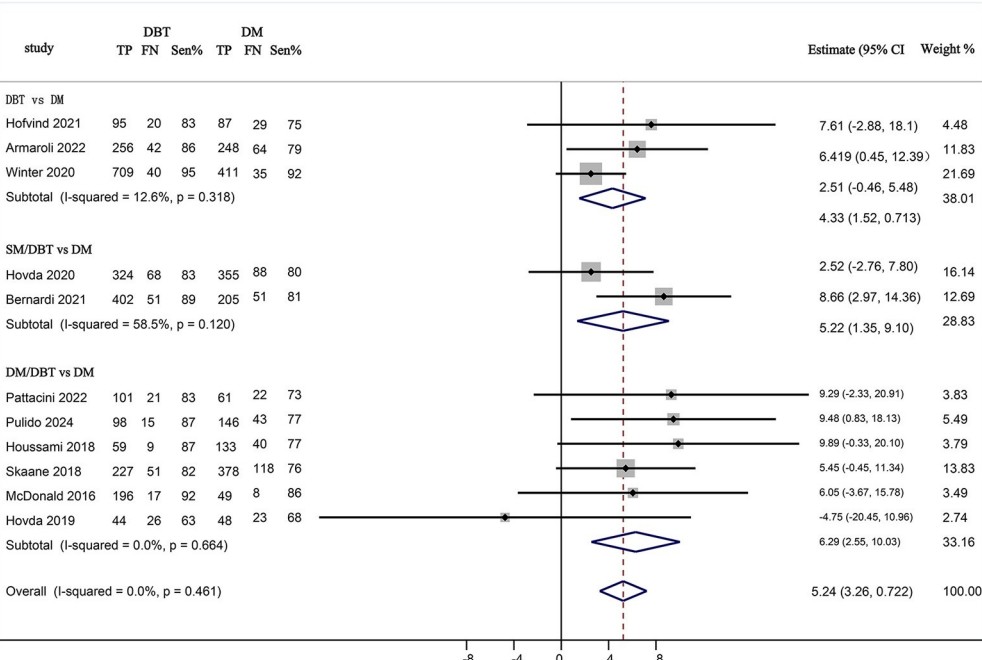

**Fig 5. Forest plots of screening sensitivities of DBT alone, DM/DBT, and SM/DBT compared to DM alone.** TP = true-positive, FN = false-negative, Sen% = screening sensitivities.

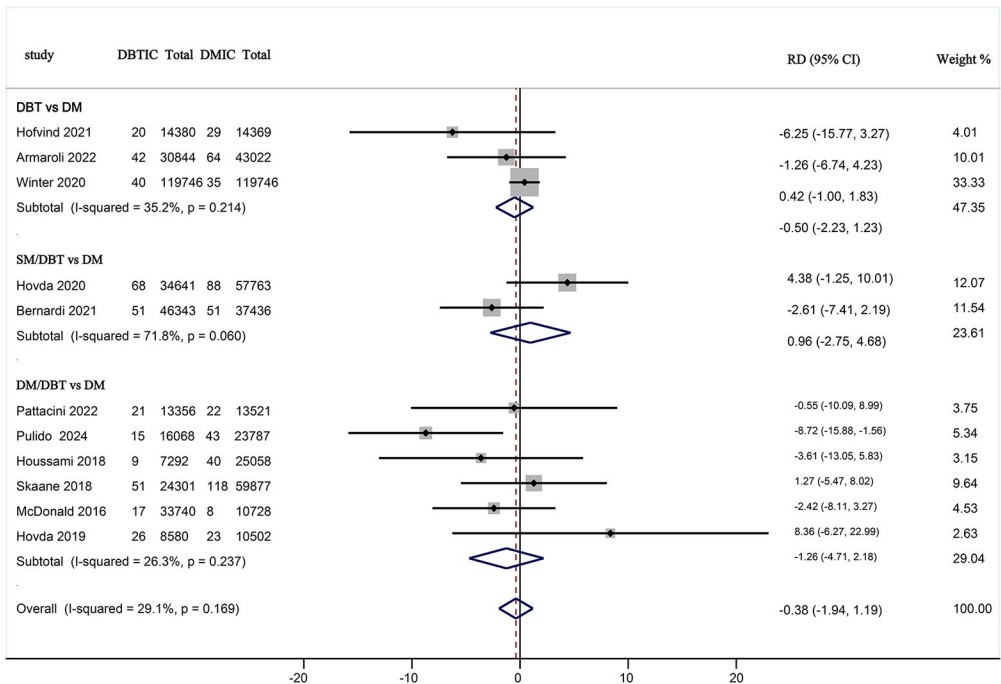

**Fig 6. Forest plots of ICR for DBT alone, DM/DBT, and SM/DBT compared to DM alone.** DBTIC = Interval cancer detected by DBT screening, DMIC = Interval cancer detected by DM screening, Total number of participants.

for DM alone, 521 cases of interval BC were identified. The effect of DBT (DBT alone, DM/DBT, and SM/DBT) on the incidence of interval BC was assessed in comparison to DM alone. Using a random effects model, the pooled risk difference was estimated at RD = -0.380% (95% CI: -0.19, 1.19, $I^2$ = 29.1%<50%). The results indicated no significant difference in ICR between the DBT screening group and those screened with DM alone (**Fig 6**). To compare various interventions, three studies were included that compared digital DBT alone with DM screening alone, encompassing a total of 339,465 subjects and detecting 230 cases of interval BC. Additionally, six comparative studies of DM/DBT versus DM screening were included, involving a total of 246,865 subjects and identifying 393 cases of interval BC. Furthermore, two comparative studies of SM/DBT versus DM screening were included, with a total of 176,183 participants and 258 cases of interval BC detected.

The analysis revealed that DBT alone showed no significant difference in ICR compared to DM alone, with a pooled RD = -0.5 (95% CI: -2.23, 1.23, $I^2$ = 35.2%<50%). There was no difference in the incidence of interval BC between the use of DM/DBT and DM RD = -1.26% (95% CI: -4.71, 2.18, $I^2$ = 26.3%<50%). Similarly, there was no difference in interval BC incidence when comparing SM/DBT with DM RD = 0.96% (95% CI: -2.75, 4.68, $I^2$ = 71.8%< 75%). The sources of heterogeneity in the intervention subgroups could not be explored, as the original study did not provide data that would allow for a heterogeneity analysis. In conclusion, the impact of DBT alone, DM/DBT, and SM/DBT on ICR, when compared to DM screening alone, was not significantly different.

### 3.6. Interval breast cancer characteristics

Excluding studies that did not report the characteristics of interval BC and screening BC [23,28,30,32], (**Table 1**) summarizes the use of DBT. The histopathological characteristics of 2,511 cases of screening BC and 627 cases of interval BC were examined. The results indicated

**Table 1. Characteristics of interval versus screening breast cancer.**

| Variable | | Interval BC | Screening BC | Total | P value |
|---|---|---|---|---|---|
| **Cancer histology** | IDC | 184 (77%) | 432 (75.9%) | 616 (76.2%) | P = 0.15 |
| | ILC | 31 (13%) | 83 (14.6%) | 114 (14.1%) | |
| | Tubular | 3 (1.3%) | 20 (3.5%) | 23 (2.8%) | |
| | Other | 21 (8.8%) | 34 (6%) | 55 (6.8%) | |
| | Not report | 376 | 1513 | | |
| **Grade (invasive)** | 1 | 85 (16%) | 609 (33.7%) | 694 (29.7%) | P<0.01 |
| | 2 | 237 (44.6%) | 880 (48.6%) | 1117 (47.7%) | |
| | 3 | 209 (39.4%) | 320 (17.7%) | 529 (22.6%) | |
| | NA | 67 | 97 | | |
| **Lymph node** | Negative | 227 (56.2%) | 1079 (76.6%) | 1306 (72%) | P<0.001 |
| | Positive | 177 (43.8%) | 330 (13.4%) | 507 (28%) | |
| | Not report | 50 | 104 | | |
| **Subtype** | LuminalA | 97 (42.9%) | 330 (58.9%) | 427 (54.3%) | P<0.001 |
| | LuminalB her2 negative | 72 (31.9%) | 148 (26.4%) | 220 (28%) | |
| | LuminalB her2 positive | 17 (7.5%) | 29 (5.2%) | 46 (5.9%) | |
| | HER-2 positive | 13 (5.8%) | 19 (3.4%) | 32 (4.1%) | |
| | Triple-negative | 27 (11.9%) | 34 (6.1%) | 61 (7.8%) | |
| | NA | 13 | 15 | | |
| | Not report | 473 | 1828 | | |
| **Size pT (invasive)** | ≤10mm | 77 (28.4%) | 445 (68.1%) | 522 (56.5%) | P<0.001 |
| | 10-20mm | 109 (40.2%) | 167 (25.6%) | 276 (29.9%) | |
| | >20mm | 85 (31.4%) | 41 (6.3%) | 126 (13.6%) | |
| | Unknown | 71 | 39 | | |
| | Not report | 263 | 17101 | | |

P value for comparison of the distribution between screening BC and interval BC. Not reported: Includes missing data, NA = not applicable (these data were not counted in the distribution of proportions).

that the proportion of Grade 3 tumors in interval BC was significantly higher than that in screening BC (39.4% vs 17.7%), with a statistically significant difference (p<0.01). The proportion of positive axillary lymph node metastasis in interval BC was significantly higher than that in screening BC (43.8% vs 13.4%) and the proportion of pT (<20mm) was also higher than that in screening BC (31.4% vs 6.3%) with a statistically significant difference (p<0.001). Furthermore, the proportion of the Luminal A molecular subtype in interval BC was lower than that in screening BC (42.9% vs 58.9%) this difference was statistically significant (p<0.001).

Excluding studies that failed to report the interval BC characteristics between different screening methods [23,28,30,32], (**Table 2**) summarizes the characteristics of 627 cases of interval BC. Among these cases, 242 were detected using DBT alone, DM/DBT, and SM/DBT, while 385 cases were identified using DM alone. The proportions of Grade 2 tumors, negative lymph node metastasis, molecular subtypes (Luminal B/HER2-, HER2), and pT (10–20 mm) in the DBT screening group were higher than those in the DM alone. However, these differences were not statistically significant (all p>0.05).

In conclusion, there is no significant difference in the distribution of characteristics of interval BC between the different screening methods. However, when compared to screening-detected BC, interval BC presents a higher degree of malignancy and a greater prevalence of non-luminal A subtypes. This difference is statistically significant (p<0.001).

**Table 2. Interval cancer characteristics of DBT versus DM screening.**

| Variable | | DBT | DM | Total | P value |
|---|---|---|---|---|---|
| **Cancer histology** | IDC | 81 (76.4%) | 103 (77.4%) | 184 (77%) | P = 0.276 |
| | ILC | 13 (12.3%) | 18 (13.5%) | 31 (13%) | |
| | Tubular | 3 (2.8%) | 0 | 3 (1.3%) | |
| | Other | 9 (8.5%) | 12 (9%) | 21 (8%) | |
| | Not report | 129 | 247 | | |
| **Grade (invasive)** | 1 | 31 (15.5%) | 54 (16.3%) | 85 (16%) | P = 0.192 |
| | 2 | 99 (49.5%) | 138 (41.7%) | 237 (44.6%) | |
| | 3 | 70 (35%) | 139 (42%) | 209 (39.4%) | |
| | NA | 29 | 38 | | |
| **Lymph node** | Negative | 84 (58.3%) | 143 (55%) | 227 (56.2%) | P = 0.518 |
| | Positive | 60 (41.7%) | 117 (45%) | 177 (43.8%) | |
| | Not report | 21 | 29 | | |
| **Subtype** | LuminalA | 43 (41.7%) | 54 (43.9%) | 97 (42.9%) | P = 0.958 |
| | LuminalB her2 negative | 34 (33%) | 38 (30.9%) | 72 (31.9%) | |
| | LuminalB her2 positive | 7 (6.8%) | 10 (8.1%) | 17 (7.5%) | |
| | HER-2 positive | 7 (6.8%) | 6 (4.9%) | 13 (5.8%) | |
| | Triple-negative | 12 (11.7%) | 15 (12.2%) | 27 (11.9%) | |
| | NA | 7 | 6 | | |
| | Not report | 150 | 323 | | |
| **Size pT (invasive)** | ≤10mm | 29 (28.2%) | 48 (28.6%) | 77 (28.4%) | P = 0.989 |
| | 10-20mm | 42 (40.8%) | 67 (39.9%) | 109 (40.2%) | |
| | >20mm | 32 (31.1%) | 53 (31.5%) | 85 (31.4%) | |
| | Unknown | 26 | 45 | | |
| | Not report | 103 | 160 | | |

P value for comparison of the distribution of interval cancer variables between DBT and DM screens. Not reported: Includes missing data, NA = not applicable (these data were not counted in the distribution of proportions).

## 4. Discussion

DBT is an innovative technology that addresses the detection limitations of DM caused by overlapping breast tissue, thereby enhancing both sensitivity and specificity [34]. This meta-analysis included eleven comparative studies (n = 762,513), revealing that the pooled sensitivity of DBT was 86% (95% CI: 81, 90), which exceeds that of DM alone at 80% (95% CI: 76, 84). Although the specificities were comparable, with both methods showing 96% (95% CI: 95, 98) and 96% (95% CI: 95, 97), respectively, the advantages of DBT in enhancing sensitivity should not be overlooked. In addition, a univariate meta-regression analysis was conducted, revealing that intervention methods and study type may serve as potential sources of heterogeneity.

While ICR has been demonstrated to be effective in evaluating the performance of screening methods [5], the question of whether DBT can reduce ICR, thereby confirming its effectiveness in BC screening, remains an area requiring further investigation [35]. In contrast to previous meta-analyses [36], this study offers the advantage of not only assessing the use of DBT in isolation but also examining its application in conjunction with other screening modalities (DM/DBT and SM/DBT) to evaluate the sensitivity of BC screening and ICR indicators. Furthermore, we analyzed the histopathological characteristics of interval BC and screen-detected breast cancer across different screening methods and found no significant differences between the two. However, when compared to screened BC, the degree of malignancy was found to be higher, with a predominance of non-Luminal A (non-LumA) type cancers

(p<0.001). These findings are consistent with the results reported in studies [37,38]. DBT, DM/DBT and SM/DBT demonstrated higher screening sensitivity compared to DM alone. However, the impact on ICR was similar across these modalities. This finding suggests that DBT screening may be susceptible to overdiagnosis. A population-based study in the United States estimates that approximately 1 in 7 BC detected through biannual screening in women aged 50 to 74 may be overdiagnosed, a figure that is lower than suggested by some other studies. This discrepancy indicates a potential overestimation in those studies [39]. The risk of BC overdiagnosis remains an uncertain factor in contemporary screening programs, highlighting the need for further studies with longer follow-up periods and additional rounds of screening to accurately assess the extent of overdiagnosis.

Previous controversies surrounding BC screening programs tend to emphasize the potential adverse effects while neglecting the significant benefits of advanced imaging technology. Current research underscores the advantages of DBT as a complementary screening tool for women with dense breast tissue. A multi-center RCT demonstrated that in women with extremely dense breasts, the combination of SM/DBT increased the detection rate of invasive BC by 48% relative to DM screening alone [40]. Further studies suggest that DBT [41], along with ultrasound [42] or MRI [43], should be utilized for supplementary screening in women with dense breasts. These compelling findings advocate a transition in BC screening from conventional DM screening to a more personalized approach that considers individual risk factors.

However, the potential of DBT to reduce ICR remains a contentious issue, with current evidence indicating variability in ICR among women with dense breasts who have undergone DBT screening. Consequently, understanding the distribution of interval BC about varying breast densities will be crucial for future assessments of the impact of DBT on ICR. Furthermore, a multicenter RCT conducted in the Netherlands revealed that supplementary MRI screening in women with dense breasts and negative DM results led to a significantly lower ICR compared to DM alone [44] and another study demonstrated that the implementation of adjunctive ultrasound in women with dense breasts can enhance screening sensitivity and potentially reduce ICR [45]. Consequently, future studies should consider performing network meta-analyses to assess the effects of various BC screening methods, on interval BC and their efficacy across different breast densities and risk factors. However, this will require more intricate methodologies and further investigation.

This meta-analysis has several limitations. First, it included only three RCTs, and some prospective studies may also be classified as retrospective reader studies. Second, when assessing the diagnostic efficacy of DBT, the absence of separate data from different samples limits the ability to analyze imaging abnormalities and various symptoms independently, hindering a comprehensive evaluation of DBT's screening performance. Notably, most studies on DBT's screening performance do not report data on interval BC. Third, there is within-group heterogeneity, as some original studies did not provide analyzable data, preventing exploration of the sources of this heterogeneity. Fourth, the small number of BC cases in this meta-analysis, coupled with a large sample size in the trials, was insufficient to establish a significant effect of DBT on ICR. Therefore, larger-scale screening or pooled analyses of BC cases occurring after DBT and DM screenings are necessary, and RCT or prospective cohort studies are necessary to assess the impact of screening interventions on ICR.

## 5. Conclusion

DBT significantly enhances the sensitivity of BC screening, yet its effect on the ICR remains uncertain. This highlights the need for further scientific research to fully understand both the

effectiveness and potential risks of DBT in practical applications. Considering the risk of increased overdiagnosis associated with DBT, future studies should prioritize the development of personalized screening strategies to optimize the benefits of BC screening.

## Supporting information

**S1 File. Search strategies.**
(DOCX)

**S2 File. PRISMA checklist.**
(DOCX)

**S1 Table. Study characteristics and patient demographics.**
(DOCX)

**S2 Table. Meta-regression analysis.**
(DOCX)

**S3 Table. All articles.**
(DOCX)

**S4 Table. Original study data.**
(DOCX)

**S1 Fig. Quality assessment of included studies.** (A)Risk of bias and applicability concerns graph, (B)Risk of bias and applicability concerns summary.
(TIF)

## Author Contributions

**Data curation:** Xuewen Liu, Ting Yang.

**Supervision:** Juan Yao.

**Writing – original draft:** Xuewen Liu.

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
