## [Decision Letter · Decision Letter 0]

30 Sep 2024

PONE-D-24-37386Impact of Digital Breast Tomosynthesis on Screening Performance and Interval Cancer Rates Compared to Digital Mammography: A Meta-AnalysisPLOS ONE

Dear Dr. Yao,

Thank you for submitting your manuscript to PLOS ONE. After careful consideration, we feel that it has merit but does not fully meet PLOS ONE’s publication criteria as it currently stands. Therefore, we invite you to submit a revised version of the manuscript that addresses the points raised during the review process.

We look forward to receiving your revised manuscript.

Kind regards,

Le Zhang

Academic Editor

PLOS ONE

Journal Requirements:

2. We note that you have referenced (unpublished) on page 6, which has currently not yet been accepted for publication. Please remove this from your References and amend this to state in the body of your manuscript: (ie “Bewick et al. [Unpublished]”) as detailed online in our guide for authors

3. As required by our policy on Data Availability, please ensure your manuscript or supplementary information includes the following:

Reviewers' comments:

Reviewer's Responses to Questions

**Comments to the Author**

1. Is the manuscript technically sound, and do the data support the conclusions?

Reviewer #1: Partly

Reviewer #2: Partly

2. Has the statistical analysis been performed appropriately and rigorously? 

Reviewer #1: I Don't Know

Reviewer #2: Yes

3. Have the authors made all data underlying the findings in their manuscript fully available?

Reviewer #1: No

Reviewer #2: Yes

4. Is the manuscript presented in an intelligible fashion and written in standard English?

Reviewer #1: No

Reviewer #2: Yes

5. Review Comments to the Author

Reviewer #1: -line 55, pp.3, Interval cancer should be Interval BC, in order to keep the context consistent.

-What is the definition of ICR? Why ICR is important in BC screening?

-line71, pp.4: What's "RCT" short for here? This is problematic: “DBT detected 70% more breast cancers than DM”. Do you want to say "DBT detected 70% more breast cancers, which was higher than that of DM"?

-What is the current status of review papers on BC screening methods and their impact on ICR? It looks this paper lacks the related work.

-Section 2.2: what are the key words for searching?

-Table 1 should include more information about the data of each study, such as where did the cohort come from, what's the screening performance reported in each study. It's important to see if the studies used the same or different cohort(s) and what the performance was.

-Which studies were included in Table 2 and Table 3, respectively?

-It's unclear how the numbers were calculated. For example, in line 294, pp.17, "The sensitivity of DBT was higher than that of DM alone, with values of 86% (95% CI: 81-90) and 80% (95% CI: 76-84)", how many studies that reported the sensitivity of DBT was higher than that of DM alone? How did you calculate the numbers of 86% and 80%?

There are some typos or grammatical errors:

-line 58, pp.3: be more invasiveness -> be more invasive

-line 59, pp.4: types of cancers detected -> types of detected cancers

Overall, the current presentation of the paper is unclear and the conclusion is vague to me.

Reviewer #2: The figure2 and figure3 in the paper are not clear.

What other conclusions can be drawn from sifting through this data?

It is suggested that the content should be added to explain the concrete support of the data to the conclusion

6. PLOS authors have the option to publish the peer review history of their article (what does this mean?). If published, this will include your full peer review and any attached files.

Reviewer #1: No

Reviewer #2: No

---

## [Author Response · Author response to Decision Letter 0]

7 Nov 2024

Dear Dr. Le Zhang and Reviewers,

On behalf of my co-authors, I would like to extend our sincere gratitude for the opportunity to revise our manuscript. We deeply appreciate the editor and reviewers for their constructive comments and suggestions.

We have studied the reviewer’s comments carefully and made revisions marked in pink in the paper. We have tried our best to revise our manuscript according to the comments. Please find the revised version attached ‘Revised Manuscript with Track Changes’, which we would like to submit for your consideration.

First, we have addressed the journal's additional requirements as follows:

1）We have revised the manuscript to comply with the style guidelines of PLOS ONE. 

2）Our article does not reference unpublished works; Specifically, on page 6, section 2.3, line 118, we state, "Incomplete and unpublished research reports" listed as part of the exclusion criteria. Consequently, unpublished articles are not incorporated into our study.

3）In ‘S2 Table All articles’ we provide all studies identified in the literature search, including those excluded from the analyses. In ‘S3 Table. Original study data’ the data used for the analyses are provided. 

4）To evaluate the risk of bias in the included studies, we employed the modified QUADAS-2 scale, as detailed in section 3.2 of the manuscript. We offer a detailed overview of the risk of bias for each study in ‘S2 Fig Quality assessment of included studies.’

We hope that our revised manuscript meets journal requirements.

Second, The principal changes made in the manuscript, along with our responses to the reviewers’ comments, are detailed below:

Reviewer #1:

1)Comment: (line 55, pp.3, Interval cancer should be Interval BC)

Response: Thank you for your constructive comments and for making my article more rigorous. We have made corrections according to the Reviewer’s comments. Line 62, pp.4, modified the term 'Interval cancer' to 'Interval BC' to maintain consistency within the context. 

2)Comment: (What is the definition of ICR? Why is ICR important in BC screening?) 

Response: Thank you very much for raising this significant point and for drawing our attention to the article's absence of a definition for ICR and its importance in the context of breast cancer screening. I would like to express my gratitude for your insightful question, which has not only inspired me but also enhanced the overall rigor of the article. We have now added this information to the Introduction section. You can find the definition of ICR and its impact on BC screening on pages 3-4, lines 53-61. Thank you for your contribution.

3)Comment: (line71, pp.4: What's "RCT" short for here? )

Response: We appreciate your inquiry. We apologize for any confusion caused by the incorrect use of abbreviations. In this article, RCT refers to randomized controlled trials. We have made the necessary correction to replace 'RCT' with 'randomized controlled trial' on page 4, line 77.

4)Comment: (“DBT detected 70% more breast cancers than DM” Do you want to say "DBT detected 70% more breast cancers, which was higher than that of DM"?) 

Response: We sincerely apologize for our previous inaccurate expression. The cited literature indicates that the cancer detection rate for DBT combined with DM was 70% higher than that for DM alone (101 cases compared to 61 cases, yielding a relative detection rate 1.7). This finding suggests that DBT and DM are more effective for cancer detection than DM alone. Line 78, pp.4, We have revised our statement to read: 'DBT combined with DM detects 70% more breast cancers than DM.'

5)Comment: (What is the current status of review papers on BC screening methods and their impact on ICR?) 

Response: Thank you for your thoughtful question; we truly value your engagement. After careful consideration, we have made several additions to our manuscript to address your concerns. In the Introduction section, on pages 3-4 (lines 53-61), we have included detailed information regarding the effects of ICR on BC screening. Additionally, in the Discussion section on pages 18-19 (lines 338-356), we've expanded on the current status of BC screening, along with an in-depth analysis of the impact of ICR in this area.

6)Comment: (Section 2.2: what are the keywords for searching?)

Response: Thank you for bringing this issue to our attention. We sincerely apologize for the omission of search keywords in Section 2.2. We have now supplemented the search keywords in Section 2.2 and included additional detailed keywords in the supporting information, 'S1 File: Search Strategies.'

7)Comment: (Table 1 should include more information about the data of each study, such as where the cohort comes from, and what the screening performance reported in each study. It's important to see if the studies used the same or different cohort(s) and what the performance was.)

Response: I appreciate your advice, which has been very enlightening. It is indeed true, as the Reviewer suggested, that additional content should be included in Table 1. Following your recommendation, we have added the following information: 'Interval cancer ascertainment and follow-up (same for comparison?), Period of screening (same or different cohort), Screen reading, and RR of Interval cancer rate (95% CI)'. Due to the increased length and width of the table, we have uploaded Table 1 as an image in the supplementary information to better display its content. Detailed data can be found in 'S1 Fig Study Characteristics and Patient Demographics'.

8)Comment: (Which studies were included in Table 2 and Table 3, respectively?)

Response: We sincerely appreciate your valuable feedback. The data sources in Tables 2 and 3 comprise studies that reported the pathological characteristics of interval BC in the meta-analysis, while studies that did not provide these characteristics were excluded. The studies enumerated in Table 2 comprise those by Hofvind (2021), Hovda (2020), Pulido-Carmona (2024), Armaroli (2022), Skaane (2018), and Hovda (2019). Table 3, in concordance, incorporates an identical selection of studies: Hofvind (2021), Hovda (2020), Pulido-Carmona (2024), Armaroli (2022), Skaane (2018), and Hovda (2019). We sincerely apologize for our oversight regarding the excluded studies not specifically listed, and for any inconvenience this may have caused. We have revised the manuscript to address this issue on page 14, section 3.6.

9)Comment: (How many studies reported the sensitivity of DBT was higher than that of DM alone? How did you calculate the numbers of 86% and 80%?)

Response: We express our sincere gratitude for your review of our article and the questions you posed. We have given careful consideration to your question, and here are our responses. According to the formulas for calculating sensitivity (TP / (TP + FN)) and specificity (TN / (TN + FP)), Stata software automatically performs these calculations. The software utilized a random effects model to conduct a meta-analysis, producing summary values for both sensitivity and specificity. The results are displayed in Figure 2, which features forest plots for DBT (A) and DM (B). This figure shows that DBT has a higher sensitivity of 86% (95% CI: 81, 90) compared to DM, which has a sensitivity of 80% (95% CI: 76, 84). Additionally, Figure 2 indicates that there are 10 studies in which DBT demonstrates greater sensitivity than DM.

10)Comment: (Some typos or grammatical errors)

Response: First, we would like to thank you for your thorough review. We apologize for the oversight regarding some typographical and grammatical errors. We have made the following corrections: 

- On Page 4, line 65, "be more invasiveness" has been changed to "be more invasive." 

- On Page 4, line 66, "types of cancers detected" has been revised to "types of detected cancers." 

Thank you for your understanding.

11)Comment: (The paper is unclear, and the conclusion is vague.)

Response: We extend our gratitude for the rigorous review and the insightful comments you have provided regarding our manuscript. We sincerely apologize for any lack of clarity or precision in our initial submission, which may have led to ambiguity in the conclusions presented. We acknowledge the challenges this may have posed to you during your reading of the article and deeply regret any inconvenience caused. We have taken your comments to heart and have endeavored to address these issues by refining the manuscript. We have refined the article and revised the Conclusion section on page 20 (lines 375-380). We hope that this will enhance the accuracy of the paper and provide a clearer conclusion to the article.

We are grateful for your contribution to the enhancement of the article's clarity and comprehensiveness, which will greatly benefit our readers in their understanding of this critical topic.

Reviewer #2:

1)Comment: (The figure2 and figure3 in the paper are not clear.)

Response: We sincerely apologize for any inconvenience caused by the unclear images in figures2 and figure3. To address this issue, we utilized professional, journal-approved tools to enhance the clarity of these images and ensure that they meet the required standards. We have re-uploaded Figures 2 and 3 and hope that the revisions we have made meet your expectations.

2)Comment: (What other conclusions can be drawn from sifting through figure2 and figure3? It is suggested that the content should be added to explain the concrete support of the data to the conclusion)

Response: Thank you for reviewing our manuscript and providing constructive feedback, which has been instrumental in enhancing its quality. We appreciate your input and completely agree with your thoughts. We’ve made some additions and improvements to Section 3.3 to enhance its clarity and depth. You can find all the specific changes we made listed in Section 3.3 on pages 9 to 11. Your feedback has been incredibly helpful, and we’re grateful for your contribution!

We trust that these revisions will enhance the clarity and rigor of the article, thereby supporting its conclusions more effectively.

We want to express our gratitude for your invaluable insights, which have significantly enhanced the quality of our manuscript. We hope that the revisions we have implemented meet your expectations.

We look forward to your response.

Kind regards,

Juan Yao

E-mail: 13139929005@163.com

---

## [Editor Report · Decision Letter 1]

26 Nov 2024

Impact of Digital Breast Tomosynthesis on Screening Performance and Interval Cancer Rates Compared to Digital Mammography: A Meta-Analysis

PONE-D-24-37386R1

Dear Dr. Yao,

We’re pleased to inform you that your manuscript has been judged scientifically suitable for publication and will be formally accepted for publication once it meets all outstanding technical requirements.

Kind regards,

Le Zhang

Academic Editor

PLOS ONE
---

## [Editor Report · Acceptance letter]

5 Dec 2024

PONE-D-24-37386R1 

PLOS ONE

Dear Dr. Yao, 

I'm pleased to inform you that your manuscript has been deemed suitable for publication in PLOS ONE. Congratulations! Your manuscript is now being handed over to our production team.

Kind regards, 

on behalf of

Dr. Le Zhang 

Academic Editor

PLOS ONE